# Pulmonary Fibroelastotic Remodelling Revisited

**DOI:** 10.3390/cells10061362

**Published:** 2021-06-01

**Authors:** Peter Braubach, Christopher Werlein, Stijn E. Verleden, Isabell Maerzke, Jens Gottlieb, Gregor Warnecke, Sabine Dettmer, Florian Laenger, Danny Jonigk

**Affiliations:** 1Institute for Pathology, Hannover Medical School, 30625 Hannover, Germany; Werlein.Christopher@mh-hannover.de (C.W.); maerzke.isabell@mh-hannover.de (I.M.); laenger.florian@mh-hannover.de (F.L.); jonigk.danny@mh-hannover.de (D.J.); 2German Center for Lung Research (DZL), Biomedical Research in Endstage and Obstructive Lung Disease Hannover (BREATH), 30625 Hannover, Germany; gottlieb.jens@mh-hannover.de (J.G.); Warnecke.Gregor@mh-hannover.de (G.W.); dettmer.sabine@mh-hannover.de (S.D.); 3Department of ASTARC, University of Antwerp, 2610 Wilrijk, Belgium; Stijn.Verleden@uantwerpen.be; 4Respiratory Division, University Hospital Antwerp, 2650 Edegem, Belgium; 5Division of Thoracic and Vasculature Surgery, University Hospital Antwerp, 2650 Edegem, Belgium; 6Department of Respiratory Medicine, Hannover Medical School, 30625 Hannover, Germany; 7Department of Cardiothoracic and Transplant and Vascular Surgery, Hannover Medical School, 30625 Hannover, Germany; 8Department of Radiology, Hannover Medical School, 30625 Hannover, Germany

**Keywords:** interstitial fibrosis, lung, alveolar fibroelastosis

## Abstract

Pulmonary fibroelastotic remodelling occurs within a broad spectrum of diseases with vastly divergent outcomes. So far, no comprehensive terminology has been established to adequately address and distinguish histomorphological and clinical entities. We aimed to describe the range of fibroelastotic changes and define stringent histological criteria. Furthermore, we wanted to clarify the corresponding terminology in order to distinguish clinically relevant variants of pulmonary fibroelastotic remodelling. We revisited pulmonary specimens with fibroelastotic remodelling sampled during the last ten years at a large European lung transplant centre. Consensus-based definitions of specific variants of fibroelastotic changes were developed on the basis of well-defined cases and applied. Systematic evaluation was performed in a steps-wise algorithm, first identifying the fulcrum of the respective lesions, and then assessing the morphological changes, their distribution and the features of the adjacent parenchyma. We defined typical alveolar fibro-elastosis as collagenous effacement of the alveolar spaces with accompanying hyper-elastosis of the remodelled and paucicellular alveolar walls, independent of the underlying disease in 45 cases. Clinically, this pattern could be seen in (idiopathic) pleuroparenchymal fibro-elastosis, interstitial lung disease with concomitant alveolar fibro-elastosis, following hematopoietic stem cell and lung transplantation, autoimmune disease, radio-/chemotherapy, and pulmonary apical caps. Novel in-transit and activity stages of fibroelastotic remodelling were identified. For the first time, we present a comprehensive definition of fibroelastotic remodelling, its anatomic distribution, and clinical associations, thereby providing a basis for stringent patient stratification and prediction of outcome.

## 1. Introduction

Fibroelastotic remodelling (FER) is a common morphological injury pattern occurring in a number of different clinical settings [1,2,3]. Within the broad scope of FER, distinct patterns with slightly diverging histopathological features and clinical phenotypes are recognized. A rare entity of interstitial lung disease (ILD) with a predominant pattern of FER, Pleuroparenchymal fibro-elastosis (PPFE), was first implemented by Frankel et al. in 2004 [4] and was officially recognized by the American Thoracic Society in 2013 [5]. PPFE often occurs idiopathically without a known trigger (iPPFE), but there are also cases of secondary PPFE linked to autoimmune disorders (AID). Moreover, PPFE-like patterns of FER have also been observed following radio- and/or chemotherapy [6] as well as after lung (LTX) and hematopoietic stem cell transplantation (HSCT) [7,8,9,10,11]. All these manifestations share a rather poor prognosis and similar histological features—a fibrous obliteration of the alveolar airspaces associated with preservation and hyper-elastosis of the embedded alveolar septa [6,12]. This histologic pattern of remodelling has been addressed by a variety of terms, including airway-cantered fibro-elastosis [13], intra-alveolar fibrosis with elastosis [14] or (intra)alveolar fibro-elastosis (IAFE, AFE) [15,16]. Moreover, it has also been recognized that the so-called “pulmonary apical caps” (PAC) of the upper lobes share strikingly similar histologic features with PPFE, but usually remain asymptomatic and are often discovered incidentally in resection or autopsy specimens [17]. A comprehensive review of the clinical manifestations of FER and the associated clinical settings is given by Chua et al. (2019) [18] which also attempts to separate clinical and pathological terminology; however, no clearly applicable minimum requirements for the histopathological diagnosis of AFE have been defined thus far. This lack of clear separation and of clinical features and nomenclature on the one hand and morphological terminology, on the other hand hampers the scientific and clinical dialogue. For instance, the clinical term PPFE is still used widely in current studies to describe the histologic pattern of FER [3,19]. In this study, we have reviewed cases with well-defined AFE pattern from the archive of Europe’s largest lung-transplant centre and systematically analysed the histological features and distribution of the FER in order to i. define stringent histological criteria, ii. clarify the corresponding terminology and iii. to distinguish relevant variants of FER.

## 2. Materials and Methods

We identified pulmonary specimens with FER in the archives of the Institute of Pathology at Hannover Medical School, sampled within the last ten years. To avoid conflicting terms we used stringent definitions for clinical and histological nomenclature (see Table 1).

For histologic evaluation, sections with approximately 1 µm thickness were cut from the formalin-fixed and paraffin-embedded archival tissue blocks and stained with haematoxylin and eosin (HE), periodic acid-Schiff (PAS) and elastic van Gieson (EvG) stains.

To define a basis for systematic evaluation, 14 PPFE and PPFE-like as well as 12 PAC cases were systematically reviewed to develop a reference catalogue of morphological features. Subsequently, all included cases included were systematically evaluated and the observed morphological features methodically catalogued.

We defined typical AFE as i. collagenous effacement of the alveolar spaces with ii. accompanying hyperelastosis of the remodelled and paucicellular alveolar walls characterized by fourfold thickening of the elastic layer when compared with normal alveoli and iii. at least 4 connected alveoli showing these changes.

The systematic evaluation was performed by two trained pulmonary pathologists (P.B. and C.W. or I.M.) in four steps on a dual-observer transmitted light microscope (Olympus BX43) equipped with 2×–40× objective lenses. First, areas of typical AFE were identified, their size (in n alveoli) estimated and the morphological features within the typical AFE regions assessed. In steps two and three, morphological features directly adjacent to the typical AFE and their respective distribution in the anatomical compartments of the lung were evaluated. Finally, the cases were re-evaluated for pathological changes in the lung but not in direct spatial association with the AFE. The consensus of both observers was recorded for future analysis. In case of disagreement cases were discussed with two consulting pulmonary pathologists (D.J. and F.L.). The study was in accordance with the regulation of the ethics committee of the Hannover Medical School (ethics vote no. 2050−2013).

The last available in vivo CT before resection was assessed by a single trained thoracic radiologist to assess if radiologic changes were completely conclusive with the histological changes, were partly conclusive or inconclusive. 

## 3. Results

We evaluated a total of 45 cases. In detail, the PPFE collective consisted of 31 cases. Of these, 6 were iPPFE and 25 had PPFE-like pulmonary fibrosis due to other causes: with underlying AID (*n* = 2), after radio-/chemotherapy (*n* = 2), chronic allograft dysfunction (CLAD) after LTX (*n* = 12) and following HSCT (*n* = 7). Two cases had AFE pattern as a minor companion pattern in other ILD.

The PAC collective consisted of samples from a total of 14 patients; of these, 12 were incidental PAC (surgery due to unrelated indications) and 2 patients had undergone primary resection due to an unclear pulmonary apical mass.

The patients were between 9 and 61 years old (See Table 2). Detailed patient information and the diagnoses of our collective are shown in Appendix A.

After a review of representative AFE cases, we could identify different patterns within typical AFE and a range of common changes in direct proximity to the AFE lesions (Table 3).

Within areas of typical AFE, the pattern of intra-alveolar fibrosis/fibrotic obliteration was classified as coarse fibrillary if broad hyalinised bundles of collagenous fibres (usually ~2 µm in diameter) were present and as fine fibrillary if delicate, mostly curled fibres were demonstrable. The presence of anthracophages and lymphoid aggregates was noted as well as an increase in cellularity with diffuse infiltration of lymphocytes or the presence of an increased number of mesenchymal cells such as (myo) fibroblasts in the obliterated alveolar lumen (see Figure 1).

In the majority of cases (67%) both coarse and fine fibrillary fibrosis could be detected, in the other cases either only fine fibrillary or only coarse fibrillary fibrosis (20% and 13% respectively) were detectable. Features found regularly in areas of AFE were aggregates of lymphatic cells (73%), often at the leading edge of the remodelling process (see Table 4). These appear well circumscribed, organized in an organoid manner, sometimes contain specialized vessels with the appearance of highly endothelialised venules (HEV) and can be distinguished from a diffuse infiltration of the AFE lesion by lymphatic cells which can be observed in approximately 30−40% of cases. Macrophages containing phagocytosed anthracotic pigment can be detected in 53% of total cases and appear less frequently in patients of the HSCT group (14%). Typical fibroblastic foci (FF) could be detected in 11% of cases. Overall, the areas of AFE showed similar morphological characteristics in all investigated groups. PAC showed overall less cellular mesenchymal (0%) and lymphatic (7%) infiltration when compared to the PPFE and PPFE-like cases (45% and 45% respectively).

We observed and catalogued several morphological patterns in direct spatial association with regions of typical AFE. Besides structurally intact lung parenchyma, various forms of FER with either fibro-elastic expansion of alveolar septa, incomplete alveolar fibrosis, or irregularly distributed collagenous and elastic fibers could be observed (See Figure 2 and Table 3 for a comprehensive list of catalogued features).

In the majority of cases (93%) fibroelastic expansion of alveolar septa could be observed at the border of typical AFE besides a direct and abrupt transition to structurally intact alveolar parenchyma (91%). In 84% of cases, areas of incomplete fibroelastosis could be detected. Frequently, pleural fibrosis could be observed adjacent to typical AFE (79%).

Patterns spatially associated with AFE were mostly similar in all cases examined, with the exception of obliterative airway remodelling (bronchiolitis obliterans; BO), which was observed in all cases of CLAD after LTX, and fibrotic pulmonary remodelling following HSCT. BO was also present in half of the PPFE cases but not present in APC.

The compartmental anatomical distribution of the delimitable changes was categorized as subpleural, parabronchial, para-arterial and paraseptal, when AFE was found in association with the respective anatomical structures. Areas of AFE in the parenchyma not associated with the anatomical structures lined out above were classified as “centrolobular” (see Figure 3).

In PAC, the AFE pattern was always found in direct spatial association with the visceral pleura (in not-PAC cases in 93%) and only extended to other compartments in a minority of cases. In contrast to PAC, PPFE and PPFE-like disease showed AFE affection of the para-arterial (97%), paraseptal (85%) and parabronchial (74%) compartments.

Other typical histologic features of ILD such as architectural distortion, myogenic metaplasia or an NSIP pattern are rarely found in direct association with AFE lesions, even if present within the same lung.

Further radiological information was available in 38 (84%) patients. Of these 38 CT scans, 26 (68%) confirmed a main pattern of alveolar fibro-elastosis, in 10 (26%) patients this was a minor pattern and in 2 (5%) remaining patients, there was no radiologic evidence for AFE (see Figure 4 for representative images)

## 4. Discussion

FER is has been considered a rather unspecific process in a multitude of diseases for over a century, until Frankel defined FER as the morphological component of a specific form of ILD termed iPPFE. However, some of the patients investigated for their study had received chemotherapy [4] and in the following years we and others identified AFE pattern as sequelae of—amongst other injuries—radiotherapy, LTX and HSCT and also concomitant with other ILD.

In their initial study, Frankel and colleagues used a rather descriptive approach and classified histological features of PPFE without establishing formal criteria. Kusagaya et al. went on to develop criteria, which were then adopted and refined by Thüsen and colleagues in 2013. These not only include intra-alveolar fibrosis and septal elastosis but also comprise a subpleural distribution in the upper lobes with concomitant pleural fibrosis.Therefore, a clear separation of the clinical and histological entities in FER is still lacking and authors often utilize “PPFE” to describe clinical, radiological and histological presentations indiscriminately [3,19]. In addition, authors consistently point out that due to the striking differences in prognosis, PPFE and PAC have to be distinguished, even though i. the histologic patterns of both are very similar and ii. both affect the upper pulmonary compartment and iii. both differ only in some aspects of spatial distribution [16]. Depending on clinical context, manifestations of AFE areassociated with different clinical outcomes (see Figure 5). 

PAC are regarded as typically benign lesions in contrast to PPFE with a mean survival time of approximately 24 months [18]. Survival in patients with ILD and concomitant PPFE varies considerably depending on the cohort reported either following the disease trajectory of the underlying ILD or of iPPFE [20,21].

To separate the clinical from the histological presentation, the term AFE was coined, describing the typical histological pattern of collagenous effacement of the alveolar spaces with accompanying hyperelastosis of the remodelled alveolar walls [15,16]. In our present study, we provide a systematic review of cases with AFE pattern histology to comprehensively document the features, distribution and pulmonary surroundings of AFE. To this end, we have employed a pattern-based approach, assessing AFE indiscriminately of its respective manifestation.

### 4.1. Features in AFE

The AFE-defining features of intra-alveolar fibrosis and septal elastosis were present in all patient groups. However, we noticed a difference in the cellular composition with an increase of lymphatic and mesenchymal cells embedded in the AFE of approximately half PPFE and PPFE-like cases, a feature we could not observe in PAC. This might point towards different states of activity within AFE lesions. Further studies are required to determine if cellularity can be used as prognostic marker. The low cellularity of AFE lesions in ILD cases with concomitant AFE is likely due to the low number of cases included in this study and typical lymphocytic inflammation could be observed in fibrotic (non-AFE) remodelled parenchyma (see Appendix A).

Anthracophages, however, were frequently encountered in PAC, PPFE and PPFE-like cases, especially in older patients, pointing towards entrapment of otherwise innocent bystanders in the fibrotic process, unlike in other ILD, where exposure to small particles is known to be a causative agent.

The presence of FF in AFE has been pointed out in several studies. Frankel et al. reported them to be rare (4). Kusagaya et al. [22] described them as appearing in “small numbers” and Von der Thüsen et al. finally as “at most in small numbers” [23]. In our study, FF can be observed in 13% of cases. Increased detection of FF in other studies could be explained by a systematic bias of the respective authors in what they consider to qualify as FF. Unlike in UIP, groups of (myo) fibroblasts observed in AFE do not readily form classical FF with perpendicularly aligned (myo) fibroblasts, embedded in an immature, myxoid extracellular matrix and accompanied by hyperplastic type 2 pneumocytes. The presence of fully formed, typical FF should, therefore, prompt the pathologist to consider AFE concomitant to another ILD.

### 4.2. Features Surrounding AFE

Incomplete AFE is very common in close spatial proximity to AFE lesions and should possibly be interpreted as an equivalent to typical AFE in the context of the clinical setting. So far it is unclear whether incomplete AFE represents an incomplete transitional state or a premature consolidation. Emphysematous and even inconspicuous alveolar parenchyma can be often observed in direct proximity to AFE lesions, which typically expand with a “pushing border” aspect into the adjacent lung. Pleural fibrosis is common and can easily be recognized. However, it is present in only about 79% of cases and should not be considered as a mandatory criterion for establishing the diagnosis.

Pulmonary arterial sclerosis is common in AFE lesions or in their close proximity both in patients with PPFE, PPFE-like disease and PAC. Some authors have suggested pulmonary arteriolosclerosis and the resulting ventilation-perfusion disparity as a trigger of AFE-type fibrosis [24].

BO is commonly found in close proximity to AFE lesions in CLAD and following HSCT, where it has long been recognized as a defining feature of the disease. In the context of PPFE-like disease, our recent study on fibrotic airway remodelling points towards shared pathways in BO and AFE development [15].

AFOP and intraalveolar macrophage aggregates can be observed in some cases. These features have been proposed to represent a transitory step in the formation of AFE [15]. The rather infrequent detection in our cohort may be explained by a temporal bias with the majority of cases being end-stage lung disease, in which AFE lesions have already consolidated. This is in agreement with a report by Von der Thüsen et al. which described AFOP in 38% of their explanted lungs following redo transplantation [23].

### 4.3. Compartmental Distribution of AFE

In our study, AFE pattern fibrosis was most commonly found in the subpleural compartment, compatible with current literature [14]. However, when excluding PAC from the analysis, AFE pattern fibrosis is also found para-arterial and para-septal in 80% of patients and para-bronchial in two-thirds of cases. This indicates that these compartments are also commonly affected in PPFE (and PPFE-like) ILD, which has a significant impact on patient survival when compared to PAC. These findings are relevant as they indicate that AFE found in other than subpleural compartments should not preclude the diagnosis of PPFE or PPFE-like disease. Moreover, the subpleural parenchyma can only be accessed by open, but not by conventional transbronchial lung biopsy. However, relevant AFE can be detected in a sufficiently large transbronchial cryobiopsy specimen [13] because fibroelastic changes extend along bronchovascular bundles and interlobular septae. Nonetheless, data regarding the sensitivity of transbronchial biopsies for diagnosing AFE is currently not available for larger patient collectives.

## 5. Conclusions

i. Cardinal features of AFE are collagenous effacement of the alveolar spaces with accompanying hyperelastosis of remodelled alveolar walls and ii. pleural fibrosis does not represent a condition sine qua non for the diagnosis. iii. Incomplete AFE has to be considered an equivalent lesion to typical AFE, provided an appropriate clinical setting. iv. FF are not a typical feature of AFE and should raise suspicion of a concomitant lung injury pattern, such as UIP. AFE is commonly distributed along the visceral pleura, the bronchovascular bundles and the paraseptal parenchyma, and compartimental involvement can give an indication towards the identification of the underlying disease. vi. The previously proposed, step-wise progression model of AFE from initial fibrinous exudation, over macrophage-rich, insufficient resolution to fully developed AFE has possibly to be complemented by active and inactive AFE, according to the intra-alveolar cellularity in the remodelled alveolar spaces.

### Outlook

The exact definition of what makes up AFE is important, not only to help to identify the underlying diseases and therefore specific treatment options, but also to stratify patients and predict their individual outcome. Moreover, the systematic application of exact histological criteria is needed as a basis for all morpho-molecular studies in order to gain further insights into the mechanisms of pulmonary FER.

## Figures and Tables

**Figure 1 cells-10-01362-f001:**
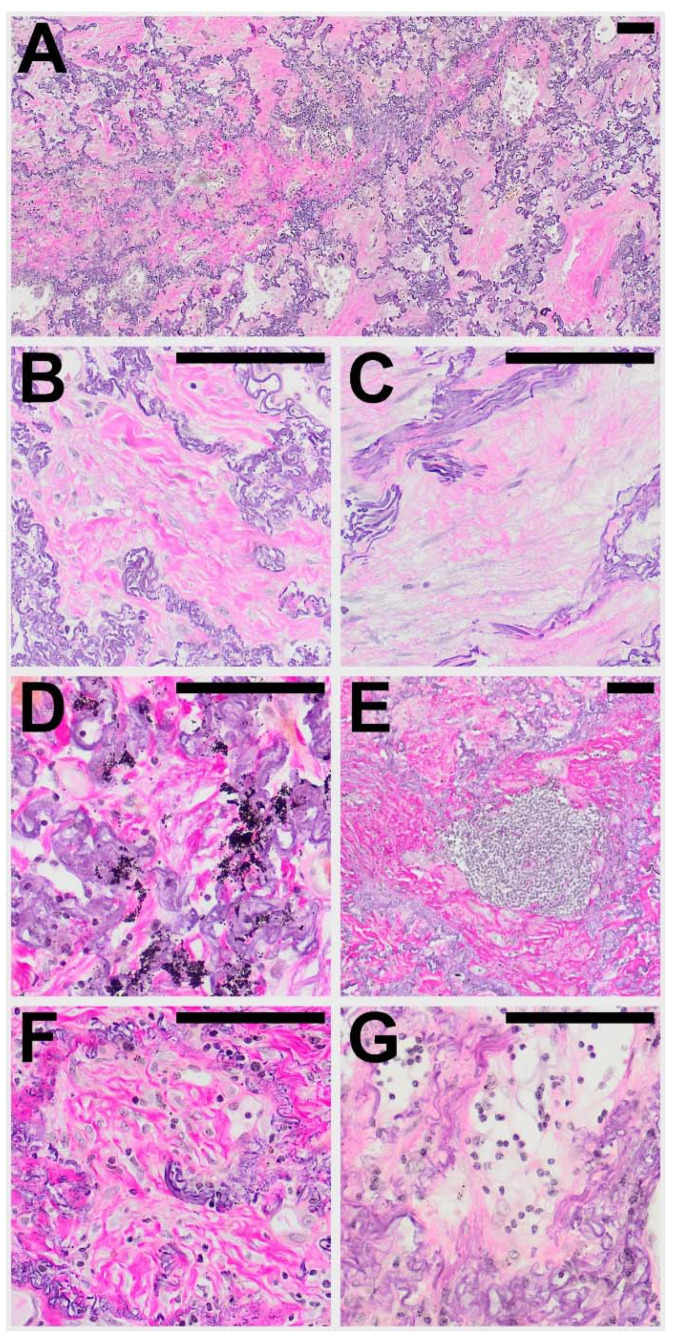
Typical histological patterns of alveolar fibroelastosis (AFE). (**A**) Typical AFE is characterized by a complete obliteration of the alveolar lumen with collagenous material with formation of either coarse (**B**) or fine (**C**) fibrils. In some cases, aggregates of macrophages containing anthracotic pigment can be observed (**D**). Lymphoid aggregates are a common finding in or at the border of AFE lesions (**E**). Increased cellularity with presence of mesenchymal cells (**F**) or lymphocytes (**G**) can be observed in the fibrotic areas to a variable degree. All images are elastic van Gieson stainings. Scale bars are 100 µm each.

**Figure 2 cells-10-01362-f002:**
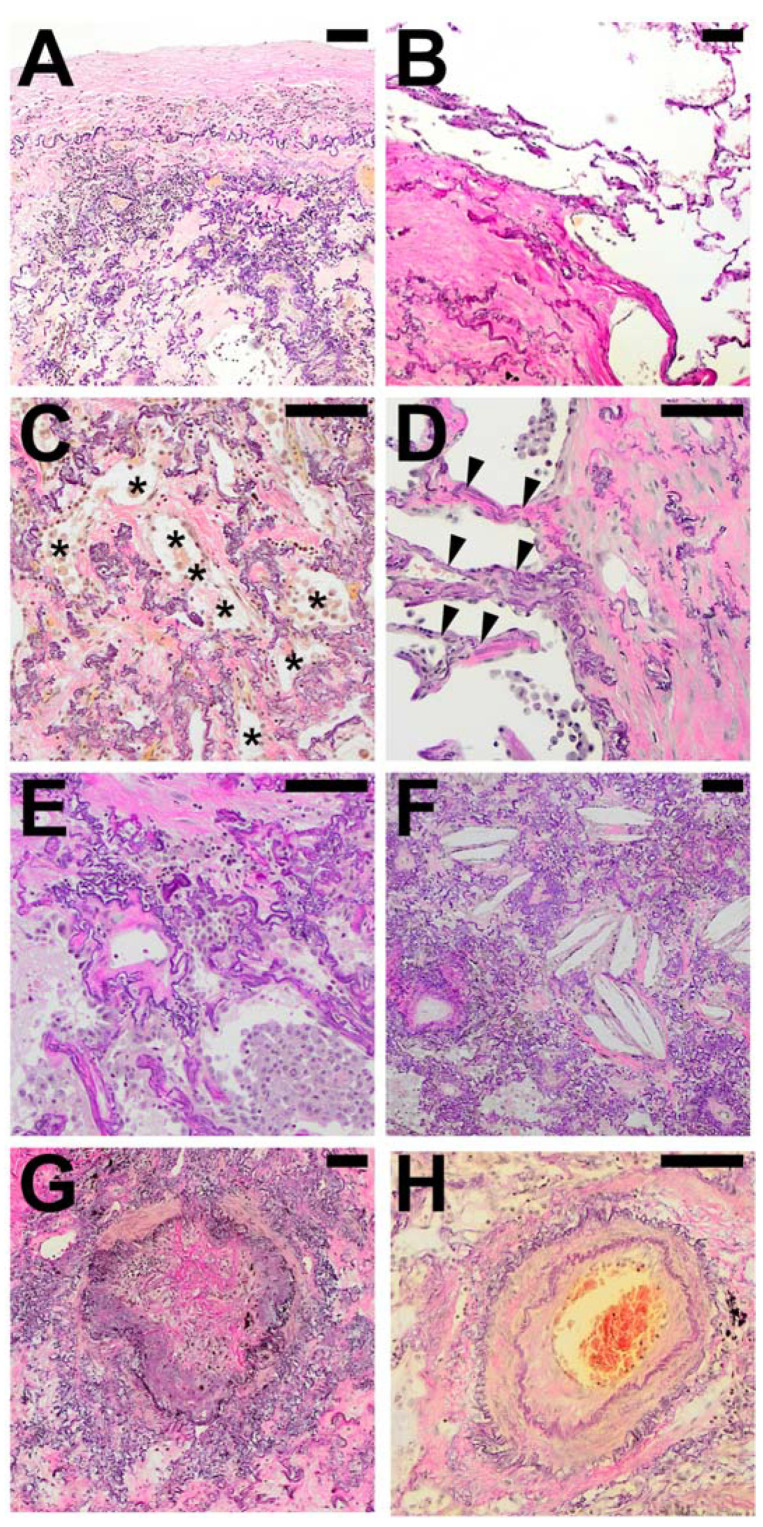
Typical histological patterns in special association with alveolar fibroelastosis (AFE). A set of typical features regularly found in spatial association with AFE: (**A**) Pronounced fibrosis of the visceral pleura. (**B**) Emphysema with an irreversible loss of alveolar septa. (**C**) Elastosis of the alveolar wall with incomplete alveolar fibrosis of the alveolar lumen with residual airspaces lined by cuboidal epithelium (*). (**D**) Fibroelastic interstitial expansion of the alveolar septa adjacent to the AFE lesion. (**E**) Aggregates of intraalveolar macrophages. (**F**) Cholesterol granulomas with multinucleated giant cells with clefts of cholesterol crystals. (**G**) Bronchiolitis obliterans with fibrous obliteration of small airways and (**H**) sclerosis of pulmonary arteries with hypertrophy of the media and intimal hyperplasia. Images are elastic van Gieson stains. Scale bars are 100 µm each.

**Figure 3 cells-10-01362-f003:**
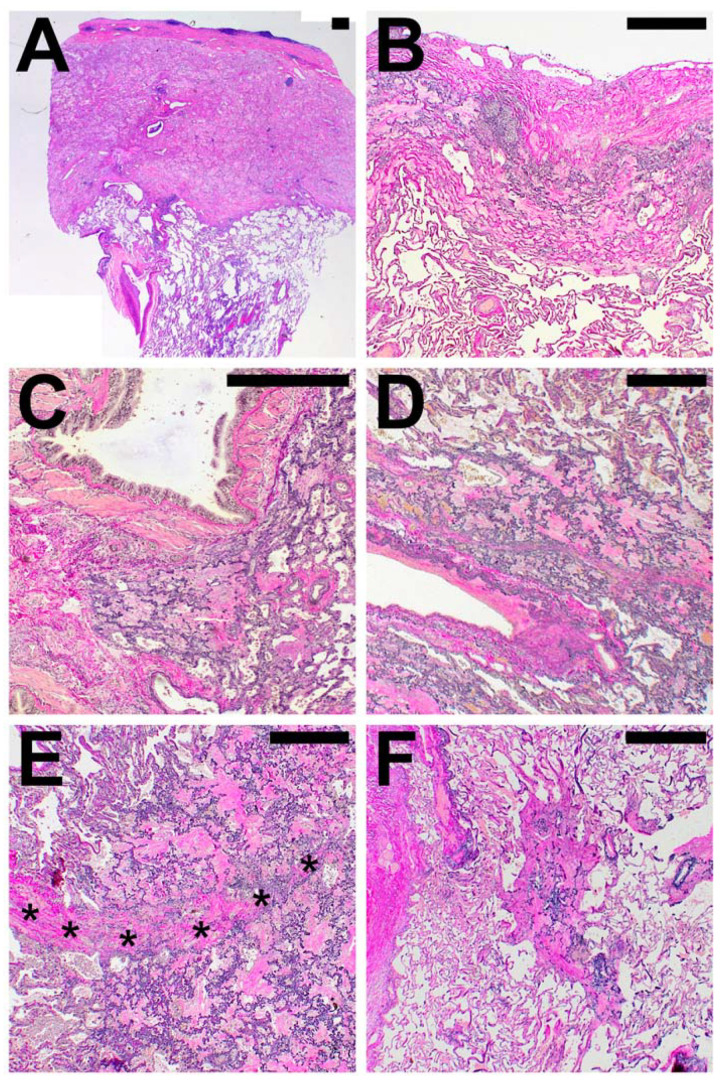
Compartmental distribution of alveolar fibroelastosis (AFE). AFE is commonly found in the subpleural parenchyma (**A**,**B**), in parabronchial (**C**) and paravascular (**D**) distribution and along interlobular septa (*, **E**). When not in association with these structures, we classified the localization as centrolobular (**F**). Images are haematoxylin and eosin (**A**) and elastic van Gieson (**B**–**F**) stains. Scale bars are 500 µm each.

**Figure 4 cells-10-01362-f004:**
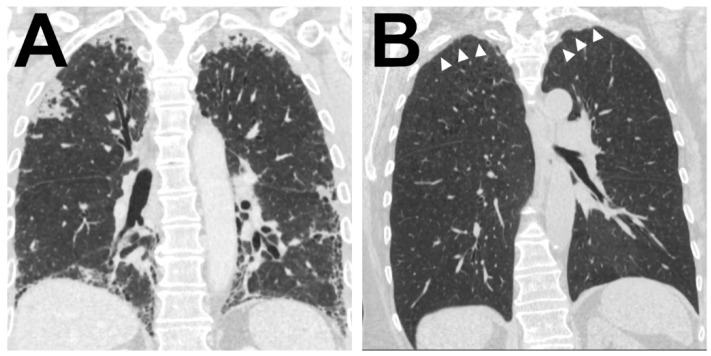
Radiological correlate of AFE Illustration of the two typical radiological patterns accompanying a histologic AFE diagnosis. (**A**). Radiological PPFE pattern with typical (sub)pleural distribution of fibrosis. (**B**) Apical cap.

**Figure 5 cells-10-01362-f005:**
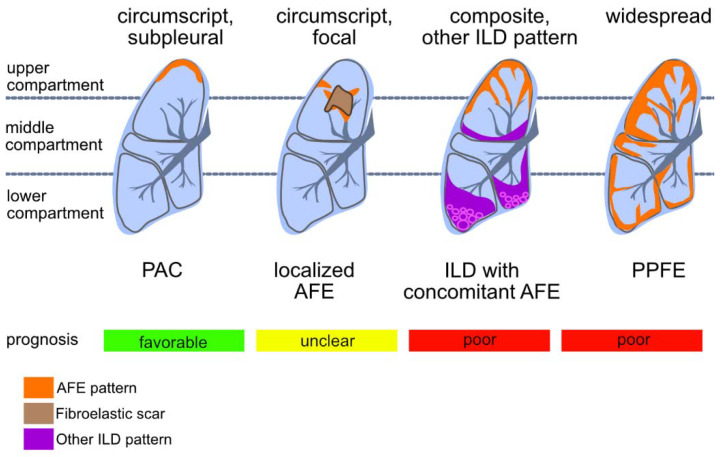
Patterns of compartmental distribution of alveolar fibroelastosis (AFE): AFE is a pattern defined by the typical fibrous obliteration of the alveolar airspace with hyper-elastosis of the preserved alveolar structure. Similar histologic patterns can be observed in a variety of diseases which are distinguishable by a characteristic distribution of the AFE pattern. When AFE is found circumscript in the subpleural parenchyma of the upper lobe without any other indication of interstitial lung disease (ILD), a prognostic favourable pulmonal apical cap (PAC) is the most likely diagnosis. Further, circumscript focal AFE can be found e.g., after radiotherapy and around (unspecific) scars of the lung parenchyma. When found in association with other ILD patterns (e.g., usual interstitial pneumonia, UIP) the prognosis of the patient may be worse than when concomitant AFE is not found. AFE as dominant pattern with subpleural, parabronchial and paraseptal distribution and accentuation in the upper compartments of the lung is indicative of pleuroparenchymal fibroelastosis (PPFE), a rare ILD with poor prognosis.

**Table 1 cells-10-01362-t001:** Terminology on fibroelastotic remodelling.

Term	Description
Fibroelastotic remodelling (FER)	Unspecific term describing matrix predominant structural changes in lung parenchyma with loss of original tissue replaced by collagenous and elastic fibers.
Alveolar fibroelastosis (AFE)	Specific histological pattern of FER characterized by collagenous effacement of the alveolar spaces with accompanying hyperelastosis of the remodelled and paucicellular alveolar walls.
Idiopathic pleuroparenchymal fibroelastosis (iPPFE)	A form of interstitial lung disease (ILD) characterized by typical clinical and radiological presentation and of unknown cause.
	Histologically predominant AFE pattern in subpleural, paraseptal and parabronchial distribution.
PPFE-like disease	FER, with clinical presentation similar to PPFE, AFE histology and known underlying disease/lung injury.
	Specifically PPFE secondary to autoimmune disease (AID),
	- alloimmune triggers (Chronic lung allograft dysfunction after lung transplantation, pulmonary fibrosis after hematopoetic stem cell transplantation) and radio-/chemotherapy etc.
ILD with concomitant AFE	Interstitial lung disease of any other type (e.g., usual interstitial pneumonia, nonspecific interstitial pneumonia) with AFE as a minor component.
Pulmonary apical cap (PAC)	Localized FER of the upper lobe with AFE histology and benign clinical course.

To avoid conflicting terms we stringently use definitions for clinical and histological nomenclature.

**Table 2 cells-10-01362-t002:** Patient collective.

Group		*n*	Age (Range)	Ratio m/f
	PAC	14	57.3 (23–77)	5/9
	iPPFE	6	52 (24–61)	3/3
PPFE-like disease	CLAD	12	45 (23–59)	4/8
	HSCT	7	24.6 (9–57)	5/2
	AID	2	44.5 (44–45)	1/1
	RCTX	2	55 (52–58)	0/2
	ILD	2	48 (47–49)	0/2

Characteristics of patient population in the investigated groups of patients with pulmonary apical caps (PAC), idiopathic pleuropulmonary fibroelastosis (iPPFE) and PPFE-like disease comprising of chronic lung allograft dysfunction (CLAD) after lung transplantation, pulmonary fibrosis after hematopoetic stem cell transplantation (HSCT), linked to autoimmune disease (AID) or radio-/chemotherapy (RCTX) and interstitial lung disease (ILD) with concomitant alveolar fibroelastosis pattern. The number of cases analysed (n), the age of the patient in years (mean and range) at the time of surgical intervention and the ratio of males (m) to females (f).

**Table 3 cells-10-01362-t003:** Histological patterns.

Pattern	Descrition
Fibroelastic interstitial expansion	Expanded alveolar septa with increased interstitial elastic and collagen fibres—often but not always—with emphysematous changes.
Normal lung parenchyma	AFE lesions can show a direct and abrupt transition to morphologically (mostly) non-remodelled lung parenchyma.
Incomplete alveolar fibrosis	Distinct hyperelastosis of the alveolar septa, similar to typical AFE. However the fibrous obliteration of the alveolar lumen is incomplete and small remnant spaces, lined by cuboidal or flat epithelium remain.
Pulmonary arterial sclerosis	Expansion of media and intima of pre-capillary pulmonary arteries.
Pleural fibrosis	Fibrotic expansion of the visceral pleura, usually with only scant cellularity.
Emphysema	Loss of alveolar septa with irreversible widening of the airspaces.
Bronchiolitis obliterans	Fibrous obliteration of small pre-terminal airways
Fibroelastotic scar	Irregularly distributed elastic and collagenous fibers without preservation of the original alveolar outlines.
Macrophage aggregates	Dominant aggregates of intraalveolar macrophages, filling upt the airspaces completely, comparable to those seen in the desquamative interstitial pneumonia (DIP) pattern.
Cholesterol granulomas	Aggregates of multinucleated macrophages with slit-like impressions of crystalline material.
Nonspecific interstitial pneumonia (NSIP)	Widening of alveolar septa. To qualify as NSIP vs. fibroelastic interstitial expansion (see above), the remodelling was required to extend uniformly throughout the lung without a gradient towards the areas of typical AFE.
Organizing pneumonia (OP), acute fibrinous organizing pneumonia (AFOP)	Aggregates of intraalveolar connective tissue (OP) or intermixed fibrin and connective tissue (AFOP) with variable, often prominent infiltration by inflammatory cells.
Myogenic metaplasia	Scattered strands of smooth muscle fibers, not associated with a bronchus or a blood vessel.
Architectural distortion	Complete loss of alveolar architecture with cystic airspace remodelling and metaplastic epithelium, as seen in usual interstitial pneumonia (UIP) pattern of lung fibrosis.

Patterns in direct spatial association with alveolar fibroelastosis (AFE) lesions were systematically evaluated in all cases.

**Table 4 cells-10-01362-t004:** Histological characteristics of alveolar fibroelastosis, its surroundings and compartmental distribution.

	Pattern	PAC	CLAD	HSCT	ILD	PPFE	All	PPFE &
								PPFE-Like
	*n* =	14	12	7	2	10	45	31
Characteristics of AFE	Fine fibrillary	100	58	57	100	90	80	71
	Coarse fibrillary	71	92	86	100	100	87	94
	Lymphatic aggregates	50	75	67	100	78	67	76
	Mesenchymal cell rich	0	50	43	50	40	31	45
	Lymphocyte rich	7	50	57	0	40	33	45
	Anthracophages	86	58	14	100	60	62	52
	Fibroblast foci	0	0	0	50	30	9	13
Spatially associated patterns	Fibroelastic interstitial expansion	93	83	100	100	100	93	94
	Normal parenchyma	100	92	71	100	90	91	87
	Incomplete alveolar fibrosis	86	75	86	100	90	84	84
	Pulmonal arterial sclerosis	64	92	71	100	90	80	87
	Pleural fibrosis	79	91	100	50	57	79	79
	Emphysema	93	83	43	50	40	69	58
	Bronchiolitis obliterans	0	92	71	0	40	44	65
	Fibroelastotic scar	21	33	14	50	50	31	35
	Macrophage aggregates	21	0	57	50	20	22	23
	Cholesterol granuloma	0	25	33	0	22	16	24
	Nonspecific interstitial pneumonia	0	0	43	0	10	9	13
	Organizing pneumonia	0	8	0	50	0	4	6
	Myogenic metaplasia	7	0	0	50	10	7	6
	Architectural distortion	0	0	0	50	10	5	7
Compartmental distribution	Subpleural	100	91	100	100	90	95	93
	Paraarterial	14	100	100	100	90	71	97
	Paraseptal	8	100	83	100	67	60	85
	Parabronchial	7	50	86	100	90	53	74
	Centrolobular	0	25	14	50	30	18	26

A total of 45 cases were systematically evaluated to assess the characteristics of typical alveolar fibroelastosis (AFE) lesions, the patterns in direct spatial association with the AFE lesion and their compartimental distribution. The cases consisted of so called “pulmonary apical caps” (PAC), chronic lung allograft dysfunction (CLAD) after lung transplantation, pulmonary fibrosis after hematopoetic stem cell transplantation (HSCT), interstitial lung disease (ILD) of other patterns with concomitant AFE and pleuroparenchymal fibroelastosis (PPFE)—either idiopathic or secondary due to autoimmune disease or radio/chemotherapy. Values are given in percentage of positive cases per group, in all 45 investigated in cases and pooled PPFE and PPFE-like cases.

## Data Availability

Datasets are freely available upon request.

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
