# Peer review of "Pulmonary Fibroelastotic Remodelling Revisited"

_cells, 2021, doi:10.3390/cells10061362_

Round 1
Reviewer 1 Report
Fibro-elastic remodeling is a term used to describe a lung injury pattern that can occur in different clinical scenarios such as idiopathic PPFE, apical caps and PPFE-like patterns observed post chemotherapy, post HSCT and post lung transplantation. The authors present a systemic evaluation of lung biopsy specimens from 45 cases with fibro-elastic remodeling (FER) from a single lung transplant centre in Europe. They have analysed the histological features and distribution of the FER in order to a) define stringent histological criteria, b) clarify the corresponding terminology and c) to distinguish relevant variants of FER associated with the different clinical settings.
This is a descriptive analysis of the histological manifestations of AFE and the associated clinical and radiological findings. The paper is generally well written and the histological images are clear.
Concerns:
- The authors define typical AFE as a) collagenous effacement of the alveolar spaces with b) accompanying hyperelastosis of the remodelled and paucicellular alveolar walls characterized by fourfold thickening of the elastic layer when compared with normal alveoli and c) at least 4 connected alveoli showing these changes. How does this definition differ from the current definition for AFE (based upon ATS guidelines 2013) ?
- Interpretation of histological findings is subject to inter-observer variability. As the histological analysis is central to this study, can the authors provide details regarding how the analyses were performed. Such as how the images were evaluated – was this by light microscopy and magnification used, was image analysis software used, how many histopathologists examined each case, were the results (% of lung affected by AFE etc) based upon a single observer or a mean value from several independent observers etc.
- The samples analysed were all obtained from lung resections (often for malignancy) and lung transplant specimens. The authors comment that transbronchial biopsy is unlikely to provide sufficient lung tissue to evaluate for subpleural involvement. As per the ATS/ERS international guidelines (2018), lung biopsy is recommended in cases of suspected PPFE. Please can this section be amended to reflect the international consensus.
- From their findings, “AFE pattern fibrosis is also found para-arterial and para-septal in 80% of patients and para-bronchial in two-thirds of cases. This indicates that these compartments are also commonly affected in PPFE (and PPFE-like) ILD,…..These findings are relevant as they indicate that AFE found in other than subpleural compartments should not preclude the diagnosis of PPFE or PPFE-like disease. ” They also comment on cellular differences in AFE between PPFE and PAC. I am not clear if the authors are proposing changes to the histological definition of AFE or histological features to help to differentiate PPFE from PAC as suggested by figure 5? Please can this be made clearer in the manuscript.
- Supplementary table 3 on page 3– please change to supplementary table 1
Author Response
Query (Q) 1.1
Fibro-elastic remodeling is a term used to describe a lung injury pattern that can occur in different clinical scenarios such as idiopathic PPFE, apical caps and PPFE-like patterns observed post chemotherapy, post HSCT and post lung transplantation. The authors present a systemic evaluation of lung biopsy specimens from 45 cases with fibro-elastic remodeling (FER) from a single lung transplant centre in Europe. They have analysed the histological features and distribution of the FER in order to a) define stringent histological criteria, b) clarify the corresponding terminology and c) to distinguish relevant variants of FER associated with the different clinical settings.
This is a descriptive analysis of the histological manifestations of AFE and the associated clinical and radiological findings. The paper is generally well written and the histological images are clear.
Response (R) 1.1
We thank Reviewer #1 for her/his kind comments and have done our best to address the concerns and suggestions put forward.
Q 1.2
The authors define typical AFE as a) collagenous effacement of the alveolar spaces with b) accompanying hyperelastosis of the remodelled and paucicellular alveolar walls characterized by fourfold thickening of the elastic layer when compared with normal alveoli and c) at least 4 connected alveoli showing these changes. How does this definition differ from the current definition for AFE (based upon ATS guidelines 2013)?
R 1.2
Unlike usual interstitial pneumonia (UIP) where clear requirements for histopathological diagnosis have been defined in the ATS guidelines (Raghu et al. 2018), the current literature only comprehensively describes the general features of fully developed – “you know it when you see it” - AFE/PPFE (e.g. Chua 2019). For our study, we have refined this definition to include a minimum boundary for describing AFE pattern fibrosis, which is especially important and essential, to institute a common basis to reliably describe changes in small tissue samples (e.g. [cryo-] biopsies). We have updated our manuscript to better address this question (lines 53 ff.)
Q 1.3
Interpretation of histological findings is subject to inter-observer variability. As the histological analysis is central to this study, can the authors provide details regarding how the analyses were performed. Such as how the images were evaluated – was this by light microscopy and magnification used, was image analysis software used, how many histopathologists examined each case, were the results (% of lung affected by AFE etc.) based upon a single observer or a mean value from several independent observers etc.
R 1.3
Histopathological cases were evaluated by at least two trained pulmonary pathologists with extensive experience in non-neoplastic lung diseases (PB and CW or PB and IM), until a firm consensus was achieved. In tie cases, consulting pathologists (DJ and FL) re-evaluated the cases, blinded to the original findings and interpretation. For evauluation a transmitted light Olympus BX43 microscope with 2x-40x Objectives (resulting in 20x-400x magnification) were used. These important aspects are now included in our revised manuscript on lines 83 ff./90 ff., and we sincerely thank Reviewer #1 for pointing them out.
Q 1.4
The samples analysed were all obtained from lung resections (often for malignancy) and lung transplant specimens. The authors comment that transbronchial biopsy is unlikely to provide sufficient lung tissue to evaluate for subpleural involvement. As per the ATS/ERS international guidelines (2018), lung biopsy is recommended in cases of suspected PPFE. Please can this section be amended to reflect the international consensus.
R 1.4
This is an important consideration. The ATS/ERS guidelines comment on PPFE, describe the underlying histologic changes and recommend a biopsy. However, the mode of biopsy is not further elaborated (Travis 2013, Raghu 2018). In our systematic evaluation we could show that in PPFE fibroelastotic scarring typically extends along bronchovascular bundles as interobular septae into the depth making the changes accessible also for sufficiently large transbronchial biopsies (e.g. cryobiopsies). This finding is in line with a study published by Kronborg-White et al. in 2018. We have amended our manuscript to better describe this on lines 310 ff.
Kronborg-White S, Ravaglia C, Dubini A, Piciucchi S, Tomassetti S, Bendstrup E, Poletti V. Cryobiopsies are diagnostic in pleuroparenchymal and airway-centered fibroelastosis. Respiratory research. 2018 Dec;19(1):1-7.
Q 1.5
From their findings, “AFE pattern fibrosis is also found para-arterial and para-septal in 80% of patients and para-bronchial in two-thirds of cases. This indicates that these compartments are also commonly affected in PPFE (and PPFE-like) ILD,…..These findings are relevant as they indicate that AFE found in other than subpleural compartments should not preclude the diagnosis of PPFE or PPFE-like disease. ” They also comment on cellular differences in AFE between PPFE and PAC. I am not clear if the authors are proposing changes to the histological definition of AFE or histological features to help to differentiate PPFE from PAC as suggested by figure 5? Please can this be made clearer in the manuscript.
R 1.5
We apologize for our imprecise wording in our original manuscript. Indeed, we have noted a systematic difference in cellularity between PPFE and PAC lesions. These changes might point to different disease activity and could aid in diagnostic decision-making. PPFE can also present with sparse inflammatory infiltrate so definite diagnosis still requires interdisciplinary correlation, especially when dealing with small biopsies/resections. These aspects are now addressed in our revised manuscript on lines 256 ff.
Q 1.6
Supplementary table 3 on page 3 – please change to supplementary table 1
R 1.6
We thank Reviewer #1 for identifying our mistake in numbering and have corrected it in our revised manuscript.
Reviewer 2 Report
In this paper, the authors present a comprehensive definition of fibro-elastotic remodeling (FER), its anatomic distribution, and clinical associations. The study is based on the material reviewed from 45 specimens with FER from one of the European centers. The manuscript provides also an overview of the manifestations of FER which is an important added value for clinicians. The manuscript is also well-organized and written. I recommend its acceptance in the present form.
Author Response
Reviewer #2
Q 2.1
In this paper, the authors present a comprehensive definition of fibro-elastotic remodeling (FER), its anatomic distribution, and clinical associations. The study is based on the material reviewed from 45 specimens with FER from one of the European centers. The manuscript provides also an overview of the manifestations of FER which is an important added value for clinicians. The manuscript is also well-organized and written. I recommend its acceptance in the present form.
R 2.1
We thank Reviewer #2 for her/his kind and encouraging words.
Reviewer 3 Report
Braubach et al. described a terminology for pulmonary FER using histology and imaging in different disease categories in their report entitled "Pulmonary fibroelastotic remodeling revisited". The manuscript is well-written. The correlation of histological findings and imaging is the highlight. However, this is limited by the available chest imaging data. The other interesting and valuable for clinicians would be prognostic value.
Comments:
- Table 4, please add total number of cases evaluated in each disease category.
- Table 4, please clarify what "all" indicated.
- Table 4, lymphocytes are normally seen in ILD given the roles of T cells, NKT cells, etc. Please explain why this characteristic was not observed in the current study.
- Figure 5, if possible, please define favorable and poor prognosis in percent of mortality. I would recommend to discuss the conclusion of figure 5 in the discussion section and perhaps provide some references.
Author Response
Reviewer #3
Q 3.1
Braubach et al. described a terminology for pulmonary FER using histology and imaging in different disease categories in their report entitled "Pulmonary fibroelastotic remodeling revisited". The manuscript is well-written. The correlation of histological findings and imaging is the highlight. However, this is limited by the available chest imaging data. The other interesting and valuable for clinicians would be prognostic value.
R 3.1
We want to thank Reviewer #3 for her/his kind comments and have addressed all suggestions and questions to the best of our abilities.
Q 3.2
Table 4, please add total number of cases evaluated in each disease category.
R 3.2
We have updated table 4 to include the numbers in each category.
Q 3.3
Table 4, please clarify what "all" indicated.
R 3.3
The “all” category includes all 45 investigated cases. We have updated the description of table 4 to adequately describe what “all” means in our revised manuscript.
Q 3.4
Table 4, lymphocytes are normally seen in ILD given the roles of T cells, NKT cells, etc. Please explain why this characteristic was not observed in the current study.
R 3.4
We thank Reviewer #3 for her/his helpful remarks and have reviewed the histology slides for the two combined ILD and PPFE cases: while a moderate lymphocytic infiltrate could be found in remodelled tissue outside of AFE, within AFE only few loosely scattered lymphocytes could be detected. Lymphocytic aggregates at the border between normal an AFE remodeled areas could readily be found, however. This might either be a feature of the “age” of the remodeled tissue or due to a sampling bias as only 2 cases with “other ILD” and accompanying AFE type changes were included in our study. Accordingly, we now discuss these limitations in our revised manuscript on lines 256 ff.
Q 3.5
Figure 5, if possible, please define favorable and poor prognosis in percent of mortality. I would recommend to discuss the conclusion of figure 5 in the discussion section and perhaps provide some references.
R 3.5
We thank Reviewer #3 for these very helpful comments and have amended the revised discussion accordingly. Pulmonary apical caps (PAC) are a frequent incidental finding, typically regarded as “benign” in clinical practice (Lagstein 2015). Large studies performed in the 1970s have described PAC in virtually all age groups, from adolescence onward. Definite data on mortality is not available, because patients presenting with these lesions are typically not followed up on. In contrast, PPFE has a reported survival rate of approximately 24 months and prognosis of ILD with additional PPFE varies between cohorts, with reported survival rates either closer to the underlying ILD or idiopathic PPFE.